# Pre-Chilling CGA Application Alleviates Chilling Injury in Tomato by Maintaining Photosynthetic Efficiency and Altering Phenylpropanoid Metabolism

**DOI:** 10.3390/plants14132026

**Published:** 2025-07-02

**Authors:** Yanmei Li, Luis A. J. Mur, Qiang Guo, Xiangnan Xu

**Affiliations:** 1Institute of Plant Nutrition and Environmental Resources, Beijing Academy of Agriculture and Forestry Sciences, No. 9 Shuguanghuayuan Midroad, Haidian District, Beijing 100097, China; lyanmei@baafs.net.cn; 2Department of Life Sciences, Penglais Campus, Aberystwyth University, Aberystwyth SY23 3DA, UK; lum@aber.ac.uk; 3Institute of Grassland, Flowers, and Ecology, Beijing Academy of Agriculture and Forestry Sciences, No. 9 Shuguanghuayuan Midroad, Haidian District, Beijing 100097, China

**Keywords:** chlorogenic acid, light energy transduction, over-winter production, phenylalanine biosynthesis, photosynthetic proteins, *Solanum lycopersicum*

## Abstract

Chilling injury can limit the productivity of tomato (*Solanum lycopersicum* L.), especially in over-wintering greenhouse. We here explored the effect of the pre-application of chlorogenic acid (CGA) in mitigating the impact of chilling on tomato. Flowering plants subjected to either chilling (15 °C/5 °C, day/night) or pre-treatment with CGA followed by chilling for 6 days and then by a two-day control recovery period were compared to plants maintained at control conditions (25 °C/18 °C, day/night). Chilling significantly affected the expression of PSII CP43 Chlorophyll Apoprotein, NAD (P) H-Quinone Oxidoreductase Subunit 5 and ATP Synthase CF1 Beta Subunit, reduced leaf Fv/Fm and increased malondialdehyde (MDA) levels, suggesting elevated oxidative stress. These correlated with reduced shoot biomass. All these aspects were mitigated by pretreatment with CGA. Transcriptomic and metabolomic co-analysis indicated that CGA also suppressed the shikimate pathway, phenylpropanoid biosynthesis and phenylalanine accumulation but enhanced cinnamic acid and indole acetate synthesis. Hence, the pre-chilling CGA protected the tomato plant from chilling injury by maintaining light energy utilization and reprograming secondary metabolism. This study describes the mechanism through which CGA pre-treatment can be used to maintain tomato productivity under chilling conditions.

## 1. Introduction

Tomato (*Solunum lycopersicum* L.) is one of the major crops produced in over-winter greenhouses in northern China. As a warm-season crop, tomato is highly sensitive to cold [1], which can severely restrict over-winter tomato production [2,3]. The optimal temperature for tomato growth ranges from 15 to 30 °C and is severely retarded when the air temperature drops below 10 °C [4]. Tomato plants are especially sensitive to temperature shock during inflorescent formation and blooming. This can lead to severe nutrient deficiencies, physiological or pathogenic disease and the formation of abnormal fruits [5]. Thus, defining means to prevent or reduce the degree of injury following chilling is an important issue to maintain high-quality greenhouse winter tomato production.

A plant experiencing low temperature stress displays a disruption in redox homeostasis, perturbed chlorophyll synthesis and mineral nutrient deficiency [6]. The shift in chlorophyll content within the photosystems diminishes energy transduction efficiency, leading to a destruction of the photosystem II reaction center, leading in turn to photoinhibition [7]. Alterations in chlorophyll apoproteins and NADH could also trigger electronic redundancy in photosystem I, inducing peroxidative reactions in the chlorophyll stroma [8]. The latter reflects on the production of reactive oxygen species (ROS) that would react with the organelle membranes [9], leading to membrane lipid peroxidation and resulting in the accumulation of malondialdehyde (MDA) [10]. Therefore, any means to counter the effects of chilling should result in the protection of photosynthetic proteins and ATP production [11]. Low temperature also diminishes the crop mineral nutrient assimilation capacity, through which the yield potential is compromised [12]. Therefore, the plant’s nutrient acquisition is also unignorable in stress alleviation.

The exogenous application of bioactive chemicals is one of the most practical methods to improve crop performance with chilling and is widely employed commercially [13,14]. Commonly used exogenous treatments mainly involve phytohormones and phytohormone-like growth regulators, including, but not limited to, salicylic acid, abscisic acid and melatonin [14,15,16]. Recently, chlorogenic acid (CGA) has emerged as a new phytohormone-like agent for crop growth regulation. The CGA is a phenylpropanoid metabolite that forms part of the shikimate pathway [17]. Plants increase endogenous CGA biosynthesis in response to changes in light quality and light duration [18] and under environmental stress. Elevated CGA levels have been linked to the increased expression of key phenylpropanoid biosynthesis genes *PAL* (phenylalanine ammonia-lyase), *C4H* (cinnamate 4-hydroxylase) and *C3H* (coumarate 3-hydroxylase) [19].

Considering mechanisms of CGA-conferred stress tolerance, this has been linked to its potent antioxidation properties preventing tissue peroxidation [20,21]. Thus, varieties of sea fennel (*Crithmum maritimum* L.) with higher CGA levels have significantly greater ROS scavenging activity [22]. The exogenous application of CGA could reduce tissue peroxidation by enhancing cellular antioxidation capacity [23] or slowing down lipid and amino acid metabolism [24]. A moderate application of CGA reduced membrane lipid peroxidation in freshly cut potatoes as well as inhibiting the activity of polyphenol oxidase to suppress the Maillard reaction (resulting in tissue browning) [24]. CGA sprays could also reduce the effects of herbicide toxicity on apple leaves resulting in lower MDA accumulation, the maintenance of leaf chlorophyll fluorescence and the increased expression of numerous antioxidation enzyme encoding genes [23].

Whilst the literature supported a role for CGA in improving tissue antioxidative responses under various conditions [25] and a pre-chilling application could protect tomato fruit from low-temperature storage injury [26], the studies assessing its effects on crop chilling resistance are still insufficient. Herein, we examine the effect of pre-chilling CGA applications on tomato plants’ cold resistance. The tomato plants were assessed for basic agronomic traits, ROS and tissue peroxidation products, combined with transcriptomic and metabolomic analysis, to explore the mechanisms through which exogenous CGA exerts its protective effects.

## 2. Material and Methods

### 2.1. Experiment Set-Up

The pot experiments were undertaken at the Institute of Plant Nutrition, Environment and Resources, Beijing Academy of Agriculture and Forestry Sciences (Beijing, China). *Solanum lycopersicum* L. var. ‘Jingfan 401’ was used, whose seeds were supplied by the Institute of Vegetable Science, Beijing Academy of Agriculture and Forestry Sciences (Beijing, China). The tomato seeds were primed in warm (~20 °C) distilled water for 30 min and then sown in standard 50-plug trays filled with commercial substrates (peat moss 75% + perlite 25%). When the seedlings reached the 2-true leaf stage, uniform individuals were selected and transplanted into 0.3 gallon pots filled with same type of substrate as one plant per pot. Then, the seedlings acclimated to pot conditions in the growth chamber under 12 h daylength with an optimal temperature range (25 °C/18 °C, day/night) until the first inflorescence appeared (about 5 weeks after sowing). Plants were fed with half-strength Hoagland solutions (Appendix A). The pots were watered on day 0 to 80% maximum substrate water holding capacity and then weighed before each watering to maintain constant water availability throughout the experiment.

This experiment was undertaken in a completely randomized design. Uniform plants with first inflorescent were selected and then moved to control environment (CK), with chilling (LL) and chilling with pre-chilling CGA application (LL-CGA). The CK group was grown under 25 °C/18 °C, day/night, the LL group was grown under 15 °C/5 °C, day/night, and the LL-CGA group was grown under 15 °C/5 °C, day/night with a CGA pre-treatment.

During an initial controlled environment phase (day 0 to day 6), the plants were grown under control conditions and weighed every other day to assess water consumption. Then, after confirming the actual evapotranspiration, all the CK and LL plants were irrigated with a nutrition solution, whilst all the LL-CGA plants were irrigated with the nutrition solution containing 0.05 g L^−1^ CGA. All of the solution was evenly spread on the surface of the growing substrate. After three irrigations (day 6), the CK plants were kept under the same conditions, while the LL and LL-CGA plants experienced 6 days’ chilling at 10 °C (day 7 to day 12), followed by 2 days of recovering growth under control conditions (day 13 to day 14). All three groups of plants only received the normal nutrition solution from day 7 to day 14. Then, at the end of day 14, the plants were harvested for assessments. The experiment details are shown in Table 1.

The 14-day experiment was conducted in growth chambers (DGX-260E, Ningbo Jiangnan Instrument Factory, Ningbo, China), with a daylength of 12 h and ~70% relative humidity. Each experiment group contained three biological replications, and each replication contained 12 pots of plants: a total of 108 pots of plants (3 groups × 3 repetitions × 12 plants).

### 2.2. The Photosynthetic Capacity, Tissue Redox Homeostasis and Membrane Damage

The leaf chlorophyll fluorescence traits were measured on day 14 using a FluorPen FP 110 (Photon Systems Instruments spol. s r.o., Drásov, Czech Republic) before harvesting the plant. Five uniform plants were selected from each experimental class, and the third fully expanded leaf was assessed. The leaf was pre-adapted to darkness for 20 min before readings were taken. Fo (minimum fluorescence in dark-adapted state), Fm (maximal fluorescence intensity under full illumination), Fv (maximal variable fluorescence, Fm–F0), ABS/RC (the photon flux absorbed by the pigments per photosystem II reaction center), TR0/RC (the maximal trapping flux with all reaction centers open per photosystem II reaction center), ET0/RC (the maximal energy flux corresponding to the electron transport beyond the primary quinone reduction per photosystem II reaction center) and DI0/RC (the maximum energy flux dissipated as heat and fluorescence per photosystem II reaction center) were measured [27]. Each measurement was taken three times.

After the fluorescence measurement, the third fully expanded leaves were sampled for tissue peroxidation products and antioxidation capacity, including malondialdehyde (MDA), hydrogen peroxide (H_2_O_2_), superoxide (O_2_^−^), total hydroxyl radical-scavenging capacity (TOC) and total antioxidation capacity (T-AOC). All analyses were performed with the Solarbio kits (Beijing Solarbio Science and Technology Co., Ltd., Beijing, China) using spectrophotometer (Thermo Scientific Multiskan SkyHigh, ThermoFisher Scientific, Waltham, MA, USA), the detailed product number given in Appendix A. Briefly, 0.1 g fresh mass was weighed into a 2 mL centrifuge tube to which 1 mL of extraction liquid was added and mixed into a slurry using a chilled mill (Bionoon—96LD, BIONNOON, Shanghai, China). This was centrifuged (21,000× *g*), with the supernatant used for later testing.

Leaf MDA content was measured according to the method described by Janero in 1990 [28,29] whereby the supernatant of the extracted solution is reacted with 0.5% thiobarbituric acid with boiling and the absorbance measured at 532 nm. H_2_O_2_ content was measured according to “Olsen’s method” [28] whereby the supernatant is mixed with 5% Ti(SO_4_)_2_ and the absorbance read at 400 nm. O_2_^−^ was assessed according to Wang and Luo’s method [30] whereby 10 mM hydroxylamine hydrochloride is added to the supernatant and then 17.3 mM sulfonamide and 8.5 mM N-naphthylethylenediamine dihydrochloride added at 37 °C. The resulting color change was read at 530 nm wavelength. Total hydroxyl radical-scavenging (TOC) ability was determined based on the strength of the Fenton reaction; and total antioxidant capacity (T-AOC) using the ferric reducing antioxidant power (FRAP) method [31]. All the measurements were taken with three biological repetitions.

### 2.3. Plant Biomass and Mineral Nutrients Accumulation

On day 14, the plants were cut just above the substrate level, and shoot and root fresh mass values were taken (LC-FA, Lichen Instruments Company, Shaoxing, China). Dry mass values were obtained by baking in a pre-heated oven (105 °C) for 30 min and drying at 75 °C for 72 h.

To obtain total nitrogen values, 0.30 g of homogenized dry sample was weighed and digested with H_2_SO_4_ and H_2_O_2_ solutions (98% H_2_SO_4_ and 95% H_2_O_2_ solution, 1:2, *v*/*v*), and then the total nitrogen content was determined using the Kjeldahl method, and the total phosphorus content measured by the Molybdenum-antimony Colorimetric method [32]. For the measurement of total potassium (K), calcium (Ca), magnesium (Mg), sulfur (S), iron (Fe), manganese (Mn), copper (Cu) and zinc (Zn), 0.30 g homogenized dry sample was weighed and carbonized on the electronic oven (600 °C) till no smoke was produced, then transferred into a muffle furnace where the sample was totally incinerated at 500 °C for 6 h. The resulting ash was dissolved in 10 mL 5% HNO_3_ solution and run through the ICP-MS (ICP-MS, Agilent 7900, Santa Clara, CA, USA). All the measurements were taken in three biological repetitions.

### 2.4. The Transcriptome and Metabolomic Analysis

The frozen samples were weighed and ground to a powder using liquid nitrogen. Samples were dispatched to Shanghai Majorbio Bio-pharm Biotechnology Co. Ltd. (Shanghai, China) for RNA sequencing and gas chromatography–mass spectrometry as relevant.

Briefly, for the transcriptomic analysis, the total RNA was extracted (TRIzol Reagent, Thermo Fisher Scientific Inc., Waltham, MA, USA) quality-checked and quantified using a 5300 Bioanalyzer (Agilent Technologies. Inc., Santa Clara, CA, USA) and the NanoDrop ND-2000 (Thermo Fisher Scientific Inc., Waltham, MA, USA), respectively. Following RNA purification and reverse transcription, the RNA-seq transcriptome library was constructed, size selected and quantified following the standard protocols (Appendix A). The transcriptomic data set was deposited in the National Center for Biotechnology Information (NCBI) database (Project number: PRJNA1223128).

The transcriptome data analysis and visualization used the Majorbio online platform (www.majorbio.com) (accessed on 1 May 2024) [33]. The clean reads from each sample were aligned to the reference genome using the HISAT2 method [34], and the mapped reads were assembled using StringTie [35]. The expression level of transcripts was calculated and then quantified for gene abundance using RNA-seq by Expectation–Maximization (RSEM) to determine differential expression genes (DEGs) [36]. Differential expression genes (DEGs) were targeted based on (1) log_2_FC| ≥ 1 in the fold change of the expression level ratio contrasting the two experimental groups and (2) having *p*-value < 0.05 in Student’s *t*-test between groups, corrected for false discovery rates (FDR) using the Benjamini–Hochberg method (BH). Functional enrichment analysis was based on Kyoto Encyclopedia of Genes and Genomes (KEGG) pathways using significantly enriched DEGs.

For the metabolomic analysis, a 0.5 g leaf sample was added to 5 mL of 0.02 mg mL^−1^ internal standard (L-2-chloro-phenylalanine, dissolved in 80% methanol solution) and 2 mL chloroform and ultrasonicated for 30 min for extraction. Then, the homogenized mixture was centrifuged, and the supernatant was dried in a stream of nitrogen, then reconstituted in 15 mg mL^−1^ methoxypyridine hydrochloride and N, O-bis (trimethylsilyl) trifluoroacetamide (BSTFA with 1% TMCS) for analysis by LC-MS (Agilent 8890B gas chromatography coupled with an Agilent 5977B mass selective detector). The details are shown in Appendix A.

The metabolomic data analysis also used the online Majorbio platform (www.majorbio.com) (accessed on 1 May 2024) [33]. The data obtained from LC-MS was uploaded to the Majorbio cloud platform. At least 80% of the metabolic features detected in any set of samples were retained from the data matrix, and the normalized data matrix was obtained based on sample mass spectrometry peaks response intensities. The variables of quality control samples with a relative standard deviation > 30% were excluded. Retained data were log_10_ transformed for subsequent analysis.

Differentially accumulating metabolites (DAMs) were targeted based on (1) having a variable in projection (VIP) score of > 1 in the orthogonal partial least squares discriminant association (OPLS-DA) model which distinguished between the experimental groups and (2) having *p*-value < 0.05 in Student’s *t*-test between groups, corrected for false discovery rates (FDR).

### 2.5. Statistical and WGCNA Analysis

The performance difference in plant growth and physiological responses between the three groups were examined through one-way ANOVA (*p* < 0.05) using JMP16 Pro (SAS Institute Inc., Cary, NC, USA). All the data were presented in mean ± standard error. The identification of DEG, Pearson correlations and weighted correlation network analysis (WGCNA) used the Majorbio online platform (www.majorbio.com) (1 May 2024).

## 3. Results

### 3.1. Assessing Cell Membrane Damage, Photosynthesis Capacity and Biomass with Chilling and CGA Effects

Chilling (LL) reduced the plant shoot dry mass by 36% compared with the control group (CK) but had no significant impact on root biomass (Table 2). As a result, LL increased the root to shoot ratio by 101%% in terms of dry mass. Pre-treating with CGA (LL-CGA) apparently increased tomato shoot dry mass accumulation by 30% compared with LL, although the changes were statistically not significant. The plant root to shoot dry mass ratio was significantly reduced by 38% in LL-CGA compared with LL.

These changes in biomass could reflect changes in photosynthesis efficiency. Leaf chlorophyll fluorescence parameters were obtained and normalized using the min-max scaling method for better visualization (Figure 1). After normalization, compared with CK, LL showed reduced leaf Fv/Fo, Fv/Fm and TRo/RC by 62%, 57.4% and 36.1% as well as a corresponding increase in DIo/RC by 443%, whilst it showed no significance in ABS/RC. Results of the CGA pre-treatment suggested that CGA could protect leaf Fv/Fm by 31.0%, based on comparisons with LL. Thus, the pre-chilling CGA application could mitigate stress-associated effects on light energy transduction in chloroplasts.

Disruption in Fv/Fm is linked to oxidative effects, and this was assessed with chilling (Table 3). Compared to CK, leaf MDA levels significantly (*p* < 0.05) increased by 48% in LL plants, whilst the TOC and T-AOC declined by 30% and 32.8%, respectively. When LL was compared with LL-CGA, the latter showed a 37% reduction in leaf MDA accumulation. TOC and T-AOC measurements also suggested that CGA treatment promoted the leaf hydroxyl radical removal and antioxidation systems, increasing the corresponding values by 19.2% and 40%.

### 3.2. Chilling and CGA Effects on Nutrient/Mineral Accumulation and Translocation

Chilling is known to affect macro and trace mineral nutrient accumulation, so it was assessed in the tomato shoots and roots, and corresponding translocation factors were also calculated (Table 4). Compared to CK, LL showed significantly reduced N, P and K levels in shoots by 12%, 24% and 19%, respectively. However, there was an obvious increase in shoot Mn concentration (by 34%), but no effect was observed on the concentrations of the other minerals. When compared to LL, LL-CGA treatments showed significantly reduced shoot Fe (by 32%) and Mn (by 33%). Next, the minerals in the tomato roots were assessed; compared with CK, LL did not have a significant impact on N, P or K levels or on most minerals. The exception was Ca and Zn, where chilling reduced their concentrations by 9.9% and 26%, respectively. CGA had striking impacts on root N levels, so the root N in LL-CGA was increased by 79% compared to CK and by 77% compared with LL. It also promoted root Mg accumulation by 13% compared with CK and 12% compared to LL. However, CGA could not counter the loss of Zn in LL compared to CK. The levels of all other minerals were unchanged in LL-CGA compared to LL and CK. When the translocation of macro and micro mineral nutrients was examined, distinctive patterns were observed. Compared to CK, in the LL group the plant root to shoot translocation factors for N, P and K were decreased by 15%, 20% and 25%, respectively; however, there were increases in Ca, Mn and Zn translocation by 20%, 37% and 38%, respectively. When LL-CGA was compared, there was a striking reduction in N translocation when related to CK and LL. However, CGA had no effect on the impact of chilling on P and K. Considering secondary and micro mineral nutrients, LL-CGA reduced the root to shoot Fe, Mn and Zn translocation factor by 40.3%, 32% and 22%, respectively.

### 3.3. Leaf Transcriptome Analysis of Chilling and CGA Effects

The underlying mechanism of CGA mitigatory effects on chilling was examined using a transcriptomic approach. Transcriptomic data obtained for each experimental treatment (Table 1) were compared by PCA (Figure 2a). This indicated that the CK transcriptomes were distinct from both the LL and LL-CGA groups across principal component (PCA) 1 (PC1, explaining 58.24% of the total variation). There were few differences between the LL and LL-CGA groups, with some overlap in the ellipses which represent the confidence intervals. The differentially expressed genes (DEGs) were identified based on two comparisons, LL vs. CK (red) and LL-CGA vs. LL (blue), and the targeted genes were compared using a Venn diagram (Figure 2b). The red circle indicates DEGs identified in LL-CGA vs. LL, whilst the blue circle represents DEGs identified in LL vs. CK. A total number of 986 DEGs were common to both comparisons. The number of DEGs solely found in LL vs. CK and LL-CGA vs. LL were 5081 and 726, respectively (Figure 2b).

The DEGs between LL vs. CK and LL-CGA vs. LL were screened for significance and then annotated based on KEGG pathways (Figure 3). Figure 3a–d suggests that LL mainly influenced cell energy metabolism, RNA translation and lipid metabolism by downregulating gene expression in these pathways; meanwhile, LL-CGA mainly influenced cell energy metabolism as indicated by the upregulation of gene expression. KEGG enrichment analysis was undertaken for these two DEG sets, and the results also suggested that the energy metabolism was the main regulation pathway through which the LL and LL-CGA impacted the plant performance. The top two pathways contributing to the significance of the KEGG enrichment analysis of the DEGs from both LL vs. CK (Figure 3e) and LL-CGA vs. LL (Figure 3f) were the photosynthesis pathway and the oxidative phosphorylation pathway, which were the two most important energy metabolism pathways in the plants.

Based on the results from KEGG enrichment analysis, DEGs linked to photosynthesis energy transduction and oxidative phosphorylation in the tomato plant cold responses are listed in Table 5. DEGs that were common to both LL vs. CK and LL vs. LL-CGA are shown in bold. This indicates that chilling significantly suppressed the genes expression of many photosystem composition proteins, NADH reductase and ATP synthases in LL plants, involving the P700 chlorophyll a, P680 chlorophyll a, P681 chlorophyll a, H^+^-translocating NADH ubiquinone reductase and F-type ATPase. Interestingly, one gene encoding ATP Synthase CF1 Alpha subunit was up-regulated.

Pre-chilling CGA application could mitigate the suppressive effects of chilling on the expression of some of the photosystem composition proteins (P680 chlorophyll a, P681 chlorophyll a, cytochrome b6/f complex), NADH reductase and ATP synthase-encoding genes (H^+^-translocating NADH ubiquinone reductase and F-type ATPase), photosystem II CP43 chlorophyll apoprotein-encoding genes, NADH-plastoquinone oxidoreductase subunit five-encoding genes and ATP synthase CF1 beta subunit-encoding genes were suppressed by chilling but significantly increased by CGA treatment. Given the biochemical importance of these genes, they could determine relative plant performance under cold stress.

To provide a deeper understanding of the relationships between the DEGs defined in LL-CGA vs. LL and the physiological and biochemical data presented in Table 1, Table 2, Table 3, Table 4 and Table 5, a WGCNA network was constructed (Figure 4). The relative correlation levels between the associated genes (374 genes for the blue module, 94 for the green module, 101 for the yellow module, 43 for the red module, 154 for brown, 450 for turquoise and one for grey) and phenotype traits of individual components were indicated by a heat map (Figure 4). This generated a summarized matrix between the genes and the growth traits, tissue peroxidation status and nutrient accumulation where distinctive correlation patterns between features are shown as distinctive modules. More detailed gene information and descriptions are provided in Appendix A.

Some of the genes involved in the photosynthetic energy transduction and ATP production showed significant up-regulation in the LL-CGA vs. LL DEG set and were classified into the green module. These showed positive and negative correlation with photosynthetic energy harvest traits and tissue oxidative stress features, respectively. The exceptions were Solyc00g500200.1 (*psbK*) and Solyc00g160280.1 (*ndhD*), which was classified into the brown module. What deserved our attention was that the genes significantly changed in both DEG sets of LL-CGA vs. LL and LL vs. CK were all classified into the green module, showing significant positive correlation with biomass, antioxidation capacity and light energy transduction efficiency traits but showing significant negative correlation with MDA. This underlies the effect of these specific genes in protecting tomato plants from biomass loss and tissue peroxidation through strengthening light energy harvesting.

### 3.4. Leaf Transcriptomic and Metabolomic Co-Analysis of Chilling—CGA Effects

To complement the transcriptomic data, an unbiased metabolomic analyses was performed on the same sample classes. The results were compared using OPLS-DA (*p* < 0.05), and the major sources of variation between the classes were identified based on VIP scores (VIP > 1) (Appendix A). Comparisons were undertaken between LL and CK and then LL and CGA. Co-analysis of DEGs and DAMs showed that the major sources of variation were the pathways involved in amino acid biosynthesis and secondary metabolism, including phenylalanine, tyrosine and tryptophan biosynthesis, phenylpropanoid biosynthesis and tryptophan metabolism (Figure 5). Figure 5 indicated the pathways metabolite names and enzymes with the corresponding EC number. The two blocks below the corresponding metabolites or genes indicate the results for comparisons between LL vs. CK (left) and LL-CGA vs. LL (right).

Figure 5 shows the changes in phenylalanine, tyrosine and tryptophan biosynthesis, phenylpropanoid biosynthesis and tryptophan metabolism. In the first pathway, compared to CK the LL class plants (block on the left) had appeared to be processing shikimate and L-arogenate to feed the accumulation of phenylalanine. The higher expression of key genes encoding chorismate synthase (4.2.3.5), aspartate-prephenate aminotransferase (2.6.1.78) and glutamate-prephenate aminotransferase (2.6.1.79) might contribute to altered metabolism. Low temperature also stimulated the transformation from chorismite to tryptophan, increasing the leaf tryptophan levels, and this could be being driven by the higher expression of genes encoding anthranilate synthase (4.1.3.27) and phenylalanine ammonia-lyase (4.2.1.20). Low temperature reduced the accumulation of tyrosine but stimulated the expression of genes encoding agrogenate dehydrogenase (1.3.1.78). Low temperature also influenced phenylpropanoid biosynthesis, reducing cinnamic acid levels, most likely to feed into the accumulation of *p*-coumaroyl quinic acid. However, less clearly, key genes encoding glycine N-acyltransferase (2.3.1.133) and 4-coumarate-CoA ligase (6.2.1.12) showed both up- and down-regulation concomitantly. Furthermore, low temperature stimulated the accumulation of serotonin and 5-hydroxyl-Indoleacetate but down-regulated the expression genes encoding L-tryptophan-pyruvate aminotransferase (2.6.1.99), aralkylamine N-acetyltransferase (2.3.1.87) and aldehyde dehydrogenase (1.2.1.3).

Following comparisons with LL (block on the right), LL-CGA plants showed increased leaf shikimate accumulation. Pre-chilling CGA application mitigated against chilling effects on the levels of shikimate, phenylalanine, tryptophan, cinnamic acid, *p*-coumaroyl quinic acid, serotonin and 5-hydroxyl-indoleacetate (Figure 5). As for the genes linked to these pathways, CGA suppressed the expression of those encoding chorismate synthase (4.2.3.5), phenylalanine ammonia-lyase (4.3.1.24), 4-coumarate-CoA ligase (6.2.1.12), glycine N-acyltransferase (2.3.1.133) but up-regulated those of L-tryptophan-pyruvate aminotransferase (2.6.1.99), amidase (3.5.1.4) and acetyl-serotonin O-methyltransferase (2.1.1.4). To summarize, CGA pre-treatment appeared to suppress phenylpropanoid biosynthesis by reducing the synthesis of key precursors but might promote the synthesis of indole acetate and melatonin.

## 4. Discussion

A major effect of lower than optimal temperatures on crop growth includes the suppression of photosynthesis, nutrient absorption and the transportation of photosynthates through source to sink relationships [37]. Simultaneously elevated secondary metabolism could also restrict cell expansion, exacerbating the loss of biomass [38]. These phenomena were observed in the LL plants of our study, whereas the LL-CGA plants countered these features that diminished plant performance when compared with the LL plants.

In this current experiment, 6 days of chilling reduced the plant biomass and mineral nutrient accumulation, which was associated with the suppression of leaf photosynthesis energy metabolism and oxidative phosphorylation. When the environment temperature is too low for the photosystems to efficiently transduce the captured light energy, the non-reduced excitation energy leads to the production of ROS [39]. This compromises the photosystems by suppressing the photosynthetic protein synthesis as well as respiratory bioenergetic generation [11]. The exogenous growth regulators could induce plant tolerance to low temperature by reducing the influence of chilling on plant photosynthesis [40]. This was in accordance with our observation that the pre-chilling CGA application maintained the leaf capacity of light capture, photosynthetic energy transduction and ATP synthesis. This allowed for the maintenance of biomass. Root to shoot dry mass ratios suggested a loss of aboveground but not root biomass (Table 2). Hence, the shoot biomass recovery induced by CGA pre-treatment was the main cause of the recovery of root to shoot dry mass ratio in LL-CGA plants. This represented an important goal for this project, which aimed to preserve the yield of tomato, where the most valuable is the fruit-bearing shoot. Thus, pre-chilling CGA treatment may be a practical way to improve the over-winter production of tomato.

The most critical recovery brought by CGA was observed on the photosynthetic energy transduction. Our transcriptomic analyses suggested mechanisms of CGA mitigatory effects against the effects of chilling stress. D1 and D2 proteins are the main components of the reaction center (RC) complex, and their regeneration directly reflects PSII working efficiency [41]. Excessive ROS is known to affect D1 and D2 protein stability, leading to photoinhibition [7], as we observed in LL plants (Figure 2a and Table 5). CP43 (PSII CP43 Chlorophyll Apoprotein) and CP47 (PSII CP47 Chlorophyll Apoprotein) are two core antennas of the PSII light harvest complex, which transports excitation energy to the RC to reduce plastoquinone [42]. Significantly, LL plants had significantly down-regulated CP43 and CP47 gene expression (Table 5). Considered together, chilling would appear to be suppressing chlorophyll light energy harvesting and transduction which would result in the significantly decline of Fv/Fm, ABS/RC and TRo/RC (Figure 2a).

With the application of CGA prior to chilling, Fv/Fm levels were maintained to a certain level, suggesting a significant improvement in PSII recovery following stress. There was an increased expression of PSII Protein K (*psbK*), CP43 and PS II P680 RC D1 Protein genes, which would also have mitigatory effects against PSII damage with chilling. The PS II Protein K is one of the main components of light harvest complex antennas, so the relative increases in CP43 and D1 protein would protect the PSII reaction center, and plastoquinone reduction would be accelerated [43]. Interestingly, CP47 did not respond to CGA pre-treatments, which could indicate the greater importance of CP43 in responding to temperature fluctuations [44].

Low temperature also suppressed the efficiency of photosystem I (PSI), as the expression of PSI P700 Chlorophyll A Apoprotein A1 and the H^+^-translocating NADH ubiquinone reductases were significantly down-regulated (Table 5). These would result in the generation of electron leakage/ROS from PSI, inducing the peroxidation reactions in chlorophyll stroma [8,9] and wider membrane lipid peroxidation, leading to MDA accumulation [10].

In this study, the pre-application of CGA significantly reduced the leaf MDA accumulation, implying an alleviation on the leaf cell membrane system damage and lipid peroxidation [39,45]. This would suggest that an important factor in oxidative stress alleviation was the promotion of between-systems photosynthetic energy transduction and the excitation energy reduction brought by CGA. The LL-CGA plants had significantly higher expressions of Cytochrome b_6_f complex (Cyt b_6_f) and H^+^-translocating NADH ubiquinone reductases (Table 5). The elevation of Cyt b_6_f number could improve electron transport from PSII to PSI, and the increase in ubiquinone reductases could accelerate the consumption of electrons in PSI, reducing electron redundancy and subsequent membrane peroxidation [8]. This would allow MDA levels in LL-CGA plant leaves to stay close to the control level (Table 2).

The transcriptomic and metabolomic co-analysis revealed the manipulation of secondary metabolism by CGA pre-treatment. These effects appeared to focus on phenylalanine metabolism and phytohormone biosynthesis which contributed to plant performance improvement (Figure 5). Under stress conditions, when the plant exhibits ROS over accumulation, augmented phenylalanine biosynthesis and metabolism would promote the synthesis of growth-retarding phytohormones, concomitantly with poorer production growth-promoting hormones [46]. The biosynthesis and metabolism of phenylalanine are critical for plant defense against environmental shock, through which plants could increase the levels of stress resistance phytohormones (e.g., salicylic acid) and develop defending structures (e.g., lignified cell walls), although there would be a cost in terms of reduced biomass accumulation [47,48].

Similarly, chilling would also activate the plant shikimate pathway and phenylpropanoid biosynthesis, leading to lignification and restricting cell wall expansion [49]. This is partially aligned with our observation. Meanwhile, chilling apparently suppressed the tryptophan to indole acetate pathway by down-regulating the key enzymes, and this could be the one of the reasons for smaller biomass in LL plants. However, CGA mitigated this effect; the plants showed a decreased synthesis and accumulation of phenylalanine and *p*-Coumaroyl CoA. Thus, the plants displayed a lesser stress response and cell injury with the CGA pre-treatment. It also promoted the expression of genes encoding L-tryptophan-pyruvate aminotransferase and increased indole acetate synthesis at gene expression level, which was in favor of higher biomass. Additionally, the increased root nitrogen accumulation should also be considered, as the additional nitrogen might become the source of indole acetate, whose biosynthesis requires amino acid (tryptophan) as basic substrate [50]. Nonetheless, a compromised P, K, Fe and Zn uptake was noticed for the LL-CGA plants as compared to the CK plants. It is critical to highlight the importance of mineral nutrients for plant overall tolerance to other environmental stresses like drought, flooding and salinity [51], which may influence the cultivation management of crops, so this point deserves further study.

Our metabolomic analyses suggested that when secondary metabolism was suppressed, the leaf accumulated higher levels of cinnamic acid. This could be due to a reduced flux of cinnamic acid to downstream metabolites, including *p*-coumaroyl quinic acid. The elevation of leaf cinnamic levels could relieve oxidative stress [52], acting as a non-enzymatic antioxidant, and the enhanced TOC and T-AOC in tomato leaf supported this hypothesis. The manipulation of cinnamic acid metabolism may be directly linked to the uptake of CGA during the chilling period, because *p*-coumaroyl quinic acid is the last precursor of CGA in its biosynthesis [21]. Thus, it is possible that although the exogenous CGA suppressed endogenous CGA biosynthesis, it acts to maintain its leaf cinnamic acid levels in the plant. This point merits further study.

## 5. Conclusions

Pre-chilling CGA application could significantly improve the chlorophyll photosynthetic energy metabolism and suppress the secondary metabolism of the post-chilling tomato plant, thereby ensuring the plant photosynthesis capacity and biomass accumulation under cold stress. CGA enhanced plant photosynthesis potential by strengthening between-systems energy transduction, NAD(P)H generation and ATP production with the expression of key enzymes. However, it promoted plant tissue development by suppressing the shikimate pathway, phenylalanine accumulation and phenylpropanoid biosynthesis, thereby increasing the antioxidant cinnamic acid and the biosynthesis of indole acetate after experiencing a chilling period. A study undertaken at a real production scale to assess the pre-chilling CGA effect in actual yields should be conducted in the future.

## Figures and Tables

**Figure 1 plants-14-02026-f001:**
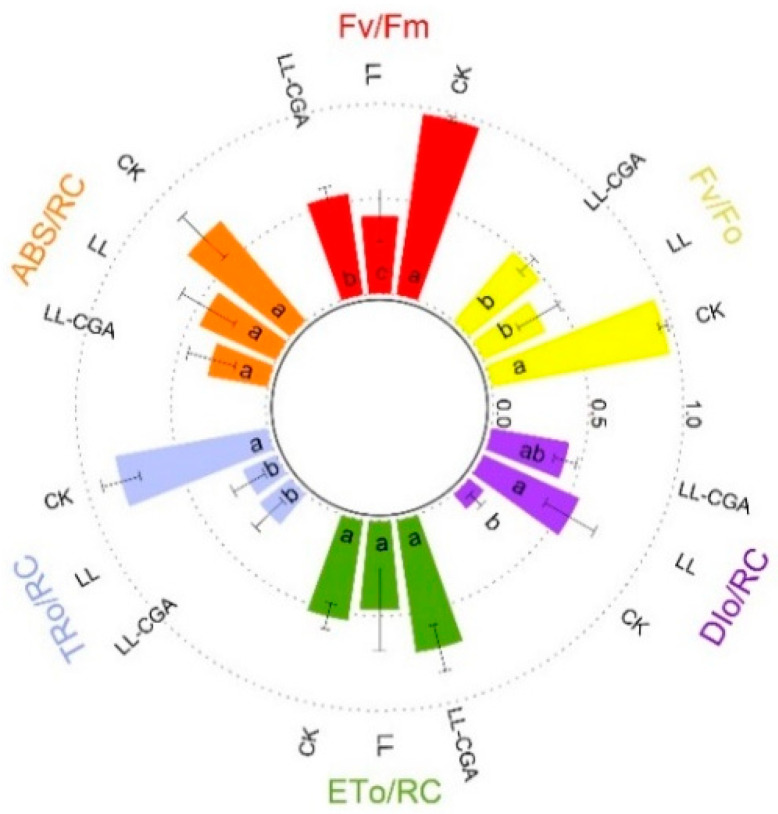
The leaf chlorophyll fluorescence parameters of tomato plants, following chilling (LL), chilling and chlorogenic acid treatments (LL-CGA) and unstressed and untreated controls (CK). The parameters assessed were Fo (minimum fluorescence in dark-adapted state), Fm (maximal fluorescence intensity under full illumination), Fv (maximal variable fluorescence, Fm–F0), ABS/RC (the photon flux uptake by the pigments per photosystem II reaction center), TR0/RC (the maximal trapping flux when all reaction centers are open per photosystem II reaction center), ET0/RC (the maximal energy flux corresponding to the electron transport beyond the primary quinone reduction per photosystem II reaction center) and DI0/RC (the maximum energy flux dissipated as heat and fluorescence per photosystem II reaction center). (*n* = 3). The different letters (a, b, and ab) indicate the different homologous groups in the multiple comparison using Student’s *T*-test.

**Figure 2 plants-14-02026-f002:**
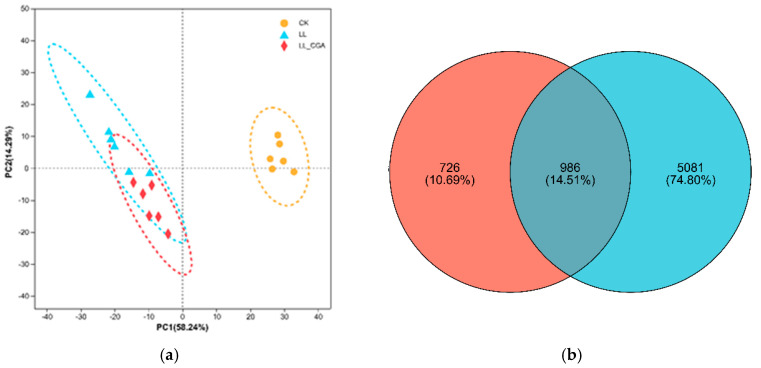
PCA analysis of the transcriptomic changes following chilling (LL, light blue data points), chilling and chlorogenic acid treatments (LL-CGA, red data points) and unstressed and untreated controls (CK, yellow data points), with dotted-line circles the confidence eclipse for each experimental class (**a**); Venn analysis of the DEG sets defined between LL-CGA vs. LL (blue circle) and LL vs. CK (red circle) (**b**).

**Figure 3 plants-14-02026-f003:**
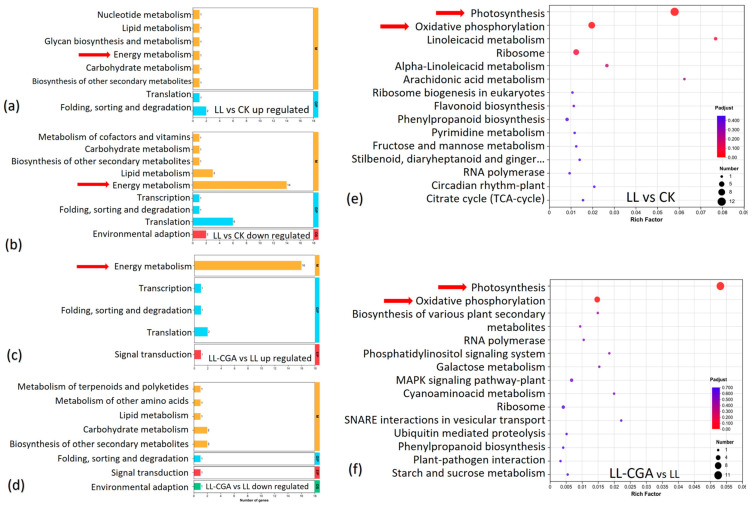
KEGG enrichment analysis of the DEGs defined by comparisons between chilling and chlorogenic acid treatments (LL-CGA) and chilling (LL) and separately between chilling (LL) and controls (CK). (**a**) KEGG pathway classification for the up-regulated genes in LL compared with CK. (**b**) KEGG pathway classification for the down-regulated genes in LL compared with CK. (**c**) KEGG pathway classification for the up-regulated genes in LL-CGA compared with LL. (**d**) KEGG pathway classification for the down-regulated genes in LL-CGA compared with LL. (**e**) Bubble chart showing the top 15 KEGG pathways enriched for the DEGs defined in LL vs. CK. (**f**) Bubble chart showing the top 15 KEGG pathways enriched for the DEGs defined in LL-CGA vs. LL. The red arrows indicate the KEGG energy metabolism pathways involving.

**Figure 4 plants-14-02026-f004:**
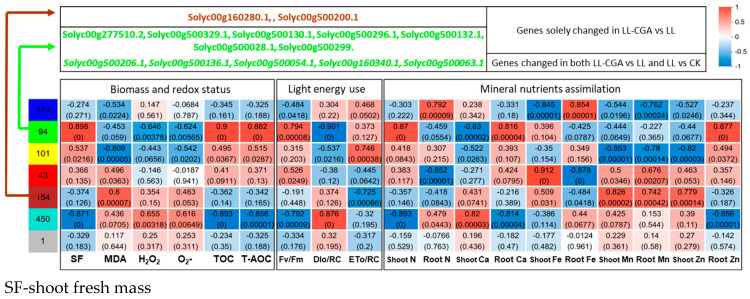
Pearson correlation matrix of the gene modules and the phenotype traits with significance in the contrasting of LL-CGA plants and LL plants (LL − CGA vs. LL). The gene modules were defined through the weighted correlation network analysis (WGCNA). The phenotypic traits are grouped into three blocks (left to right); biomass and redox status, light energy use and mineral nutrients assimilation (*n* = 3).

**Figure 5 plants-14-02026-f005:**
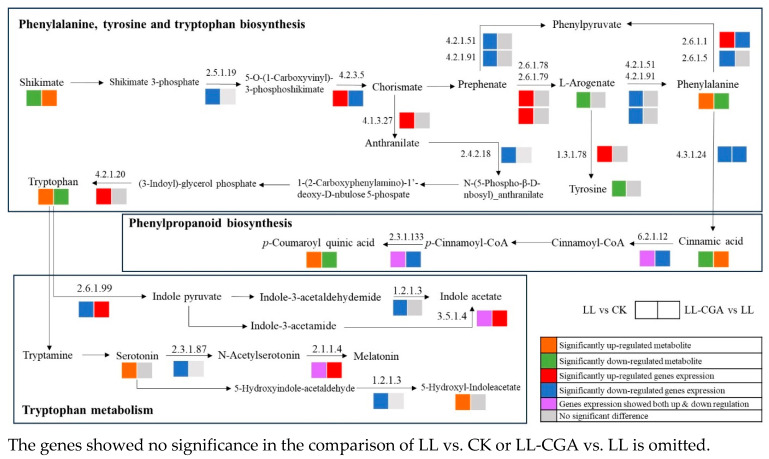
The combined transcriptomic and metabolomic results of tomato leaves from chilling (LL), chilling plus chlorogenic acid (LL-CGA) and untreated control (CK) treatments of tomato plants. The figure shows the impact of CGA pre-treatment on the tomato leaf secondary metabolism, illustrating the relation between phenylalanine, tyrosine and tryptophan biosynthesis, phenylpropanoid biosynthesis and tryptophan metabolism. The block on the left stands for the comparison result of LL vs. CK, whilst the right stands for LL-CGA vs. LL. The blocks linked to DAMs were colored orange if relatively elevated and green if down-regulated in each comparison. The blocks linked to DEGs encoding the key enzymes were colored red if relatively up-regulated and blue if down-regulated. The purple block indicated that both up- and down-regulation were detected. The grey block shows that no significant differences were observed. (*n* = 3).

**Table 1 plants-14-02026-t001:** The experiment timeline from the pre-chilling stage till the final harvest. CK—Control, LL—Chilling, LL-CGA—Chilling with pre-chilling CGA application.

	Day 0–6		Day 7–12	Day 13–14	End of Day 14
	Pre-Chilling		Chilling	Recovery	Harvest
CK	25 °C/18 °C (day/night) + 1/2 Hoagland solution	No additive	25 °C/18 °C (day/night) + 1/2 Hoagland solution	25 °C/18 °C (day/night) + 1/2 Hoagland solution	Harvest all
LL			15 °C/5 °C (day/night) + 1/2 Hoagland solution		
LL-CGA		0.05 g L^−1^ CGA			

**Table 2 plants-14-02026-t002:** The shoot and root biomass and moisture content of tomato plants. The data are presented as mean ± standard error, *n* = 3.

Treatment	Shoot DM	Root DM	Root/Shoot DM	Shoot Moisture	Root Moisture
	g	g		%	%
CK	0.22 ± 0.11 ^a^	0.03 ± 0.01 ^a^	0.13 ± 0.09 ^b^	94.1 ± 1.55 ^a^	94.2 ± 3.76 ^a^
LL	0.14 ± 0.06 ^b^	0.03 ± 0.01 ^a^	0.27 ± 0.21 ^a^	94.3 ± 3.17 ^a^	92.9 ± 1.96 ^a^
LL-CGA	0.18 ± 0.09 ^ab^	0.03 ± 0.02 ^a^	0.17 ± 0.08 ^b^	94.1 ± 2.25 ^a^	93.8 ± 3.20 ^a^
Sig.	*p* = 0.06	ns	*p* = 0.05	ns	ns

Chilling (LL), chilling and chlorogenic acid treatments (LL-CGA) and unstressed and untreated controls (CK). The different letters (a, b, and ab) indicate the different homologous groups in the multiple comparison using Student’s *T*-test. ‘Sig.’ stands for the *p*-value in ANOVA, all *p*-values smaller than 0.1 were given, whilst ‘ns’ for non-significant.

**Table 3 plants-14-02026-t003:** The leaf tissue membrane system peroxidation, ROS and antioxidation capacity. TOC—total hydroxyl radical-scavenging capacity, T-AOC—total antioxidation capacity. The data are presented as mean ± standard error, (*n* = 3).

Treatment *	MDA nmol g^−1^	H_2_O_2_ μmol g^−1^	O_2_ μmol min^−1^ g^−1^	TOC %	T-AOC U mL^−1^
CK	77.7 ± 7.78 ^b^	1.142 ± 0.44 ^a^	107 ± 0.69 ^a^	84.6 ± 4.54 ^a^	26.8 ± 0.38 ^a^
LL	115.0 ± 14.90 ^a^	1.207 ± 0.19 ^a^	108 ± 0.47 ^a^	54.6 ± 8.54 ^b^	18.0 ± 3.18 ^b^
LL-CGA	72.6 ± 14.80 ^b^	1.119 ± 0.27 ^a^	107 ± 0.71 ^a^	73.8 ± 13.30 ^a^	25.2 ± 2.36 ^a^
Sig.	***	ns	ns	***	***

Chilling (LL), chilling and chlorogenic acid treatments (LL-CGA) and unstressed and untreated controls (CK). The different letters (a and b) indicate the different homologous groups in the multiple comparison using Student’s *T*-test. ‘Sig.’ stands for the *p*-value in ANOVA, all *p*-values smaller than 0.1 were given, and ‘*’ for *p* < 0.05, ‘***’ for *p* < 0.001, whilst ‘ns’ for non-significant.

**Table 4 plants-14-02026-t004:** Macro, secondary and micro mineral nutrient accumulation in tomato plants. The data are presented as mean ± standard error, *n* = 3.

Treatment	N	P	K	Ca	Mg	Fe	Mn	Zn	Cu
	g kg^−1^	g kg^−1^	g kg^−1^	g kg^−1^	g kg^−1^	g kg^−1^	g kg^−1^	mg kg^−1^	mg kg^−1^
	Shoot								
CK	63.4 ± 1.85 ^a^	5.75 ± 0.53 ^a^	28.6 ± 1.77 ^a^	18.6 ± 0.88 ^a^	8.15 ± 0.48 ^a^	0.57 ± 0.06 ^a^	0.86 ± 0.25 ^ab^	40.1 ± 5.76 ^a^	10.1 ± 0.41 ^a^
LL	55.9 ± 1.12 ^b^	4.35 ± 0.25 ^b^	23.2 ± 1.85 ^b^	20.1 ± 0.70 ^b^	8.61 ± 0.32 ^a^	0.53 ± 0.05 ^a^	1.15 ± 0.06 ^a^	42.8 ± 8.81 ^a^	8.47 ± 0.13 ^a^
LL-CGA	56.7 ± 2.72 ^b^	4.36 ± 0.33 ^b^	25.3 ± 1.52 ^ab^	18.8 ± 1.08 ^b^	8.06 ± 0.23 ^a^	0.36 ± 0.01 ^b^	0.77 ± 0.10 ^b^	33.3 ± 3.76 ^a^	8.83 ± 1.35 ^a^
Sig.	***	***	**	*	*p* = 0.06	***	**	*p* = 0.07	*
	Root								
CK	41.9 ± 0.49 ^b^	4.47 ± 0.59 ^a^	25.3 ± 1.93 ^a^	5.85 ± 0.46 ^a^	6.48 ± 0.79 ^b^	5.19 ± 1.47 ^a^	0.92 ± 0.05 ^a^	156 ± 9.44 ^a^	10.4 ± 0.86 ^a^
LL	43.4 ± 0.62 ^b^	4.27 ± 0.43 ^a^	27.6 ± 1.39 ^a^	5.27 ± 0.25 ^b^	6.65 ± 0.06 ^b^	5.65 ± 0.46 ^a^	0.93 ± 0.17 ^a^	116 ± 11.7 ^b^	9.52 ± 0.59 ^a^
LL-CGA	77.0 ± 17.6 ^a^	4.16 ± 0.15 ^a^	28.3 ± 0.98 ^a^	5.31 ± 0.19 ^b^	7.44 ± 0.41 ^a^	6.42 ± 0.34 ^a^	0.89 ± 0.08 ^a^	117 ± 9.35 ^b^	9.74 ± 0.64 ^a^
Sig.	***	ns	*	*	*	ns	ns	***	ns
	Root to shoot translocation factor								
CK	1.51 ± 0.03 ^a^	1.29 ± 0.08 ^a^	1.13 ± 0.04 ^a^	3.18 ± 0.10 ^b^	1.27 ± 0.22 ^a^	0.12 ± 0.03 ^a^	0.92 ± 0.17 ^b^	0.26 ± 0.04 ^b^	0.98 ± 0.06 ^a^
LL	1.29 ± 0.03 ^a^	1.03 ± 0.15 ^b^	0.85 ± 0.10 ^b^	3.81 ± 0.28 ^a^	1.29 ± 0.06 ^a^	0.09 ± 0.00 ^ab^	1.26 ± 0.22 ^a^	0.36 ± 0.04 ^a^	0.89 ± 0.05 ^a^
LL-CGA	0.77 ± 0.19 ^b^	1.05 ± 0.05 ^b^	0.89 ± 0.04 ^b^	3.54 ± 0.32 ^ab^	1.08 ± 0.08 ^a^	0.06 ± 0.00 ^b^	0.86 ± 0.11 ^b^	0.28 ± 0.03 ^b^	0.90 ± 0.09 ^a^
Sig.	***	**	***	**	*p* = 0.06	**	*	*	ns

Chilling (LL), chilling and chlorogenic acid treatments (LL-CGA) and unstressed and untreated controls (CK). The different letters (a, b, and ab) indicate the different homologous groups in the multiple comparison using Student’s *T*-test. ‘Sig.’ stands for the *p*-value in ANOVA, all *p*-values smaller than 0.1 were given, and ‘*’ for *p* < 0.05, ‘**’ for *p* < 0.01, ‘***’ for *p* < 0.001, whilst ‘ns’ for non-significant.

**Table 5 plants-14-02026-t005:** Significant DEGs involved in the KEGG energy metabolism pathway determined by comparing between chilling (LL) and untreated controls (CK) and separately comparing chilling plus chlorogenic acid treatments (LL-CGA) and LL tomato plants. The relative up-/down-regulations are shown. The underlined italics indicate the same genes found in both sets of DEGs (*n* = 3).

**LL vs. CK**				
Trend	KO name	Gene name	Encoding Protein	Protein Function
down	** *psbC* **	Solyc00g500057.1, Solyc00g500209.1	**Photosystem II CP43 Chlorophyll Apoprotein**	Photosystem II (P680 chlorophyll a)
	*psbB*	** * Solyc00g500206.1, Solyc00g500136.1, Solyc00g500054.1 * **	Photosystem II CP47 Chlorophyll Apoprotein	Photosystem II (P681 chlorophyll a)
	*psaA*	Solyc00g500024.1, Solyc00g500071.1	Photosystem I P700 Chlorophyll A Apoprotein A1	Photosystem I (P700 chlorophyll a)
	*ndhA*	Solyc00g500041.1	NADH-Plastoquinone Oxidoreductase Subunit 1	NADH: ubiquinone reductase (H ^+^ -translocating)
	** *ndhF* **	Solyc00g500195.1, ***Solyc00g160340.1***	**NADH-Plastoquinone Oxidoreductase Subunit 5**	
	*atpF*	gene_Solyc00g500322.1	ATP Synthase CF0 Subunit I	F-type ATPase
	** *atpB* **	***Solyc00g500063.1**,*** Solyc00g500064.1	**ATP Synthase CF1 Beta Subunit**	
	*atpA*	Solyc00g500323.1	ATP Synthase CF1 Alpha Subunit	
up	*atpA*	Solyc00g500133.1		
**LL-CGA vs. LL**				
Trend	KO name	Gene name	Encoding Protein	Protein Function
up	*psbK*	Solyc00g277510.2, Solyc00g500329.1, Solyc00g500130.1, Solyc00g500296.1, Solyc00g500200.1	Photosystem II Protein K	Photosystem II (P681 chlorophyll a)
	** *psbC* **	** * Solyc00g500206.1, Solyc00g500136.1, Solyc00g500054.1 * **	**Photosystem II CP43 Chlorophyll Apoprotein**	Photosystem II (P680 chlorophyll a)
	*psbA*	Solyc00g500132.1	Photosystem II P680 Reaction Center D1 Protein	
	*petD*	Solyc00g500028.1	Cytochrome B6/F Complex Subunit IV	Cytochrome b6/f complex
	*ndhB*	Solyc00g500299.1	NADH-Plastoquinone Oxidoreductase Subunit 2	NADH: ubiquinone reductase (H ^+^ -translocating)
	*ndhD*	Solyc00g160280.1	NAD(P)H-Quinone Oxidoreductase Subunit 4	
	** *ndhF* **	** * Solyc00g160340.1 * **	**NAD(P)H-Quinone Oxidoreductase Subunit 5**	
	** *atpB* **	** * Solyc00g500063.1 * **	**ATP Synthase CF1 Beta Subunit**	F-type ATPase

## Data Availability

The transcriptome data is deposited in NCBI database (PRJNA1223128). Other data are available by the author on reasonable request.

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
