# Peer review of "Pre-Chilling CGA Application Alleviates Chilling Injury in Tomato by Maintaining Photosynthetic Efficiency and Altering Phenylpropanoid Metabolism"

_plants, 2025, doi:10.3390/plants14132026_

Round 1
Reviewer 1 Report
Comments and Suggestions for Authors
The article effectively demonstrates the beneficial effects of chlorogenic acid (CGA) on tomatoes exposed to cold stress; however, it does not explain how CGA is actively absorbed and translocated to the leaves, nor does it address whether it acts locally in the rhizosphere. The methodology should include a rationale for selecting this dose of CGA (0.05 g/L). The article measures high-quality variables; however, the absence of a control treatment that applies CGA without cold exposure limits the results, as it is not possible to determine whether the findings are due exclusively to the mitigation of chilling injury or if part of the response is attributable to the effects of CGA under normal conditions. The study presents an intriguing integration of transcriptomics and metabolomics; however, it does not address certain inconsistencies in metabolic pathways. For instance, while the CGA application suppresses the accumulation of phenylalanine and phenylpropanoid derivatives, the level of cinnamic acid, the immediate precursor of CGA, increases. Hence, it remains unclear why cinnamic acid accumulates if the preceding steps are inhibited. Additionally, it is necessary to discuss whether the exogenous application of CGA affects endogenous CGA biosynthesis. Another underdeveloped point is the relationship between the modulation of secondary metabolism and changes in mineral homeostasis. The article mentions an increase in nitrogen and magnesium in roots with CGA treatment, yet a reduction in translocation to the shoots. Therefore, it is not explained whether this redistribution is due to an alteration in active transport induced by CGA or to metabolic reprogramming.

Author Response
Comment 1:The article effectively demonstrates the beneficial effects of chlorogenic acid (CGA) on tomatoes exposed to cold stress; however, it does not explain how CGA is actively absorbed and translocated to the leaves, nor does it address whether it acts locally in the rhizosphere. The methodology should include a rationale for selecting this dose of CGA (0.05 g/L). The article measures high-quality variables; however, the absence of a control treatment that applies CGA without cold exposure limits the results, as it is not possible to determine whether the findings are due exclusively to the mitigation of chilling injury or if part of the response is attributable to the effects of CGA under normal conditions. The study presents an intriguing integration of transcriptomics and metabolomics; however, it does not address certain inconsistencies in metabolic pathways. For instance, while the CGA application suppresses the accumulation of phenylalanine and phenylpropanoid derivatives, the level of cinnamic acid, the immediate precursor of CGA, increases. Hence, it remains unclear why cinnamic acid accumulates if the preceding steps are inhibited. Additionally, it is necessary to discuss whether the exogenous application of CGA affects endogenous CGA biosynthesis. Another underdeveloped point is the relationship between the modulation of secondary metabolism and changes in mineral homeostasis. The article mentions an increase in nitrogen and magnesium in roots with CGA treatment, yet a reduction in translocation to the shoots. Therefore, it is not explained whether this redistribution is due to an alteration in active transport induced by CGA or to metabolic reprogramming.
Response:
Thanks for your reviewing and detailed comments. We fully understand your concern about treatments setting. The reason way we chose these three groups to evaluate the effect of CGA to tomato chilling tolerance is for directly comparing the change of plant growth and metabolism pattern when it is subjected to chilling or chilling plus CGA. No matter whether the plant would response to CGA under control condition, its physiological response with CGA is actually happened under chilling, and this is what we really care when using the pre-chilling CGA application to alleviate chilling injury. Thus, we truly believe the treatments setting could prove the CGA’s effect. But still, we do agree that the plant response under control with CGA worth a deeper study before the conclusion could be used in actual production, so we will to include this treatment in the next larger scale experiment targeting fruiting and actual yield.
Reviewer 2 Report
Comments and Suggestions for Authors
The article "Pre-Chilling Application of Chlorogenic Acid can Alleviate Chilling Injury in Tomato by Maintaining Photosynthetic Efficiency and Altering Phenylpropanoid Metabolism" presents an interesting approach to the use of chlorogenic acid for mitigating chilling stress in tomato plants. I commend the authors for their work; the manuscript is well-written, concise, and scientifically appropriate.
Below, I provide a single suggestion for revision:
Materials and Methods
Line 92: Please correct the degree Celsius symbol (e.g., ~20 °C).
Author Response
The article "Pre-Chilling Application of Chlorogenic Acid can Alleviate Chilling Injury in Tomato by Maintaining Photosynthetic Efficiency and Altering Phenylpropanoid Metabolism" presents an interesting approach to the use of chlorogenic acid for mitigating chilling stress in tomato plants. I commend the authors for their work; the manuscript is well-written, concise, and scientifically appropriate. Below, I provide a single suggestion for revision:
Materials and Methods
Line 92: Please correct the degree Celsius symbol (e.g., ~20 °C).
Response:Thanks for your suggestion, we have corrected this point and this is currently in line 93.
Reviewer 3 Report
Comments and Suggestions for Authors
Please see the attachment.

Author Response
- -Table 2: For “shoot moisture,” use lowercase “Ns” in “ns.”
-Line 285: Please introduce the abbreviation for nitrogen (N) when it is first mentioned, and use “N” consistently thereafter.
Response: Thanks for your suggestion, we have corrected these typos in the corresponding parts.
2.-Table 4: Consider whether the reduction in phosphorus after treatment with LL-CGA could negatively affect plant fertility and thus yield, given phosphorus's critical role in the reproductive phase, its involvement in DNA, etc. Also, consider whether the observed reduction in potassium in the LL-CGA treatment could impact fruit quality—particularly tomato fruit color and shelf-life - since potassium plays a key role in water regulation in fruits and throughout the plant. Could these plants also be less drought-tolerant due to lower potassium levels? Please incorporate this discussion and consider listing it as a potential limitation of the treatment.
Response: Thanks for your suggestion. We do agree that the mineral nutrients uptake may influence the overall plant resistance to environmental factors, so we added some contents discussing this point in line 528-532.
3. -The Conclusion is well written, but please include a section on future research perspectives.
Response: Thanks for your suggestion. We have added the future research perspectives targeting actual production or field trial in line 552-554.
Reviewer 4 Report
Comments and Suggestions for Authors
Plants-3680781-peer-review
Pre-Chilling Application of Chlorogenic Acid can Alleviate Chilling Injury in Tomato by Maintaining Photosynthetic Efficiency and Altering Phenylpropanoid Metabolism
Yanmei Li , Luis A.J. Mur, Qiang Guo and Xiangnan Xu
Climate change, including sudden changes in environmental temperature conditions, causes problems of plant life support, both in the greenhouse and in the open ground. Tomatoes, as year-round greenhouse plants, are especially damaged by chilling during the winter growing period. In this regard, the problem of plant resistance to temperature stressor becomes of general importance.
To improve plant stress tolerance, growth regulators of different nature, both stress hormones and antioxidant substances, are increasingly used in crop production practice. The authors of the manuscript applied pretreatment with a natural antioxidant, chlorogenic acid, as an increase in cold tolerance of tomato plants.
In the Introduction, the authors reviewed the scientific literature to characterize possible changes in plant metabolism under cold stress.
The authors evaluated the oxidative status, total antioxidant capacity, and photosynthetic activity of the third unfolded leaf. They evaluated the content of ten mineral elements: N, P, K, Ca, Mg, S, Fe, Mn, Cu, and Zn, and showed changes in their translocation under chilling and chlorogenic acid action. The authors performed transcriptomic and metabolomic analysis, showing the regulation of the expression of a large number of genes under the action of cold and chlorogenic acid application.
Figures and tables interpreting the results of the study are presented in the manuscript.
The authors showed that CGA increased the photosynthetic potential of plants by enhancing intersystem energy transfer, NAD(P)H generation and ATP production with the expression of key enzymes. When secondary metabolism was suppressed, the leaf accumulated higher levels of cinnamic acid, which could act as a non-enzymatic antioxidant, reducing oxidative stress.
Comments
1.Please clarify the cooling temperature regime of tomato plants. The night temperature in your experiment was 5 OC, where did the temperature of 10 OC come from in the abstract and text of the manuscript.
The alternation of night and day temperatures cannot be averaged because plants respond to both temperatures. The former causes the formation of a stress state, the later maintains the plant's energy status, allowing for rehabilitation (or adaptation).
2. Please enter the Latin name of the plant species Solanum lycopersicum and chlorogenic acid in Keywords.
3. In the Introduction, the possibility of changes in mineral intake should be noted.
4. A better description of mineral translocation could have been provided. For example, it is known that nitrogen absorbed by plant roots can be metabolized into amino acids and in this form enter the shoot, or serve as building material for proteins of different purposes in root cells.

Author Response
1.Please clarify the cooling temperature regime of tomato plants. The night temperature in your experiment was 5 OC, where did the temperature of 10 OC come from in the abstract and text of the manuscript. The alternation of night and day temperatures cannot be averaged because plants respond to both temperatures. The former causes the formation of a stress state, the later maintains the plant's energy status, allowing for rehabilitation (or adaptation).
Response: Thanks for your suggestion. We agreed that the description of temperature control was a little ambiguous, so we revised all the parts mention temperature setting. The control group was subjected to 25/18 °C, day/night, and the chilling group was subjected to 15/5 °C, day/night.
- Please enter the Latin name of the plant species Solanum lycopersicumand chlorogenic acid in Keywords.
Response: Thanks for your suggestion. We have added these two terms in the keywords list.
- In the Introduction, the possibility of changes in mineral intake should be noted.
Response: Thanks for your suggestion. We have added this content in line 54-56.
- A better description of mineral translocation could have been provided. For example, it is known that nitrogen absorbed by plant roots can be metabolized into amino acids and in this form enter the shoot, or serve as building material for proteins of different purposes in root cells.
Response: Thanks for your suggestion. We have added this content in the discussion part, which was in line 528-531.
Reviewer 5 Report
Comments and Suggestions for Authors
The objectives of this manuscript are duly justified with substantiated background information and relevant and up-to-date references. The Materials and Methods section is clearly written, the methodologies are well explained and supported with relevant references. The results are clearly presented and discussed. The conclusions are duly supported by the results obtained.
However, there are some observations that the authors should address. These observations are outlined below.
The first section (Introduction) has no title.
On lines 58 and 498 the Song et al., 2022 reference appears, and in the References list there are two references of Song et al., 2022, However, the authors do not clarify which corresponds in each case.
On line 124 it says “… and ͠͠ 70% humidity .” I think it should say: “… and ͠ 70% relative humidity.”
In Table 2, the result of the variable Root/shoot DM which appears for CK treatment is 0.16±0.09b, however, when dividing the values that correspond to the variables Shoot DM and shoot DM 0.22±o.11/0.03±0.01, the result is 0.13 ± 0.09 and not 0.16±0.09.
Author Response
1.The first section (Introduction) has no title.
Response: Thanks for your suggestion. The title was added.
2.On lines 58 and 498 the Song et al., 2022 reference appears, and in the References list there are two references of Song et al., 2022, However, the authors do not clarify which corresponds in each case.
Response: Thanks for your suggestion. We also noticed the citation format didn’t meet the journal requirement, so we revised all the citation and reference list into the “numbered formatting”.
3.On line 124 it says “… and ͠͠ 70% humidity .” I think it should say: “… and ͠ 70% relative humidity.”
Response: Thanks for your suggestion. We have revised this typo inline 124.
4.In Table 2, the result of the variable Root/shoot DM which appears for CK treatment is 0.16±0.09b, however, when dividing the values that correspond to the variables Shoot DM and shoot DM 0.22±o.11/0.03±0.01, the result is 0.13 ± 0.09 and not 0.16±0.09
Response: Thanks for your suggestion. We also noticed this typo, and we have revised this part.
Reviewer 6 Report
Comments and Suggestions for Authors
This manuscript has been revised. After some corrections, this manuscript is a candidate for publication.

Author Response
- Please cite all references using reference numbers and place the numbers in square brackets (“[ ]”), e.g., [1], [1–3], or [1,3]. Please refer to the following website for more information: https://www.mdpi.com/authors/references.
Response:Thanks for your suggestion. We have changed the citation formatting to align with the journal standard. All the citations were numbered and inserted, and the reference list was also changed.
- The authors are encouraged to consider the following points to meet publication standards. The manuscript contains numerous grammatical and typographical errors, which hinder readability. Look for recent studies and remove those that are more than 5 years old unless they are important. The authors must add some novel contributions to the existing model to make it useful for the contributing society. Major revisions are necessary to improve the clarity, rigor, and scientific contribution of the study.
Response:Thanks for your suggestions. We have corrected the mistakes and typos as pointed in the attached file, and we also revised the discussion part to include more useful information for further research. For the references, we replaced 2 of the old articles with one later (2022) article. For those older articles left in the list, we also rechecked the content and information they supplied, and be believed they were so valuable and still kept them. We are pretty sure we have more than 70% references published in last five years, and more than 80% references published in last 7 years.
- Format the table 1 better.
Response:Thanks for pointing this out, we have corrected table 1.